# AC/DC Electric-Field-Assisted Growth of ZnO Nanowires for Gas Discharge

**DOI:** 10.3390/ma16010108

**Published:** 2022-12-22

**Authors:** Wenming Yang, Chenjun Hao, Shengsen Zhang, Tianyang Zheng, Rong Zhu, Beiying Liu

**Affiliations:** 1School of Mechanical Engineering, University of Science and Technology Beijing, Beijing 100083, China; 2State Key Laboratory of Precision Measurement Technology and Instrument, Department of Precision Instruments, Tsinghua University, Beijing 100084, China

**Keywords:** nanowires, ZnO nanowires, ZnO, electric-field-assisted growth

## Abstract

Using ZnO nanowires as needle anodes in gas discharge is helpful for maintaining continuous discharge with a relatively low voltage. It is necessary that the ZnO nanowires are far enough apart to guarantee no electric field weakening and that the nanowire anodes are easy to assemble together with the discharging devices. An AC/DC electric-field-assisted wet chemical method is proposed in this paper. It was used to grow ZnO nanowires directly on discharging devices. The nanowires covered the whole electrode in the case in which only a DC field was applied. Moreover, the tips of the nanowires were scattered, similar to the results observed under the application of AC fields. The average distance between the tips of the highest nanowires was approximately equal to 4 μm, which almost meets the requirement of gas discharge. The research concerning growing ZnO nanowires directly on PCBs shown that, at the current time, ZnO nanowires on PCBs did not meet the requirements of gas discharge; however, in this study, the parameters regarding ZnO nanowire growth were established.

## 1. Introduction

ZnO nanowire is currently one of the most important one-dimensional nanostructures due to its wide band gap energy, biocompatibility, and ease of fabrication [1,2,3]. It is widely used in various applications such as gas sensing [4,5], photoluminescence [6,7], electromechanical oscillators [8], and the enhancement of thermoelectric modules [9]. Therefore, ZnO nanowires have received much attention to date [10]. One distinctive application is that of needle electrodes in gas discharge used as ion sources for aerosol charging [11,12,13].

Gas discharge is usually produced by a non-uniform electric field, for example, the corona that occurs between a needle and a plate or between a concentric wire and a tube [14]. If the electric field intensity in a discharge gap is sufficiently high, air or other gases within undergo electrical breakdown, and a certain amount of gas ions are produced. During this process, electrons must have enough energy to knock an electron from the gas molecules so that positive ions and free electrons are created. Until now, extremely high voltages are needed for the discharging electrodes in the majority of -dischargers, i.e., they require several kilovolts, meaning most state-of-the-art instruments are large, heavy, and expensive [15]. Simple and portable devices are necessary for many applications, such as aerosol charging and applications with other ion sources. The advantage of gas discharge with ZnO nanowires lies in the fact that the discharge can be maintained with relatively low voltages [11], which is attributed to the efficient excitonic emission of ZnO nanowires and their thin point.

Sharp nanotips grown on electrodes have already been used to enhance local electric fields, such as carbon nanotubes [16,17] and metallic [18] or semiconductor [19] nanowires. Among these different types of nanostructures, ZnO has a number of advantages over other materials, such as its efficient excitonic emissions [20]. Several approaches are applied to synthesize ZnO nanowires, including chemical vapor deposition (CVD) [21], electrochemical deposition [9], pulsed laser deposition [22], thermal evaporation [23,24], the vapor–phase method [3], and the wet chemical method [25,26]. Among these methods, the wet chemical method is favorable for directly constructing ZnO nanowires on microdevices. This method can also be assisted by an electric field to control the growing position and alignment of ZnO nanowires. Liu et al. [27], Wang et al. [28], and Yang et al. [11] used an AC or a DC electric field to control the growing position between two electrodes or on one electrode, respectively. However, for applications in gas discharge, a distinctive distribution of ZnO nanowires is required in order to achieve a low maintaining voltage of the discharge. Many experiments have shown that previous growing methods cannot achieve this goal [27,28]. In this study, we proposed a new electric-field assisting mode that used both AC and DC electric fields for the growth of ZnO nanowires to meet the requirement for gas discharge. Moreover, the requirements for nanowire electrodes in gas discharge were provided according to the calculation of the electric field distribution of the microelectrodes. In this case, ZnO nanowires grow on discharging devices and can be used for dischargers using nanowires as anodes.

## 2. Materials and Methods

### 2.1. Method of Applying AC and CD Electric Fields

In gas discharge applications, ZnO nanowires are used as one electrode. The electric field around their tips needs to be significantly enhanced so as to maintain a continuous discharge with a relatively low voltage. Therefore, the average distance between the tips of ZnO nanowires should be sufficiently large, i.e., they need to be sparsely distributed on the surfaces of the electrodes.

In order to directly grow ZnO nanowires on discharging devices, we used the electric-field-assisted wet chemical method. Previous studies have shown an aligned and dense distribution of ZnO nanowires forms under the application of a DC field. In addition, a dispersed distribution that only grows between two electrodes was obtained with the application of an AC field [7,8,11,23]. In line with these concepts, a dispersed distribution of ZnO nanowires and growth on the entire electrode was attempted using the wet chemical method with the application of both AC and DC electric fields. The electric field application method during growth is shown in Figure 1. An AC electric field was produced between one electrode and the substrate, and a DC electric field was maintained between another electrode and the substrate.

### 2.2. Method of Growing ZnO Nanowires on Microelectrodes

We first grew the ZnO nanowires on a pair of microelectrodes using the field application method shown in Figure 1. The shape of the microelectrodes used in the experiments is shown in Figure 2. During the fabrication of the dischargers, the microelectrodes and the ZnO nanowires thereon were transferred to the devices, so the dimensional parameters influenced the distribution of the electric field in the dischargers. Appropriate electrode dimensional parameters are helpful for maintaining a relatively low voltage. Therefore, we chose the simple shape shown in Figure 2, but with varying dimensional parameters, in order to determine an optimum. The dimensional parameters considered are shown in Figure 3, and their values are given in Table 1.

Ansys software was used to calculate the electric field distribution around the fingers of the microelectrodes when a voltage of 500 V was applied between the anodes and a grounded electrode. The distance between them was set as 50 μm. The electric field intensities along the path shown in Figure 3 were used to compare the microelectrodes with different dimensional parameters. Figure 4 provides the results. Because the maximum electric field intensity determines the maintaining voltage of the discharge in a discharger, this value for each parameter combination is also provided in Figure 5. It can be seen that electrodes with the Type 1 shape obtained the highest electric field intensity around one finger. Furthermore, the electric field distribution for the electrodes with more than four fingers was calculated. Figure 5 shows the corresponding results for electrodes with six and 12 fingers. It is obvious that multiple fingers on electrodes contribute to a larger discharge current, but too many fingers on electrodes result in a lower field intensity near the middle tips. As a result, electrodes with less than six fingers were used in this work.

The microelectrodes shown in Figure 2 were manufactured using a top-down method. Their fabrication process and the growth of ZnO nanowires involved the following steps: (1) A N-doped silicon wafer with low resistivity of 0.008~0.02 Ω·cm was prepared. This wafer served as a substrate in the following steps; (2) a thin layer of SiO_2_ was deposited on the silicon substrate by thermal oxidation; (3) a Cr/Au film was sputtered onto the SiO_2_ layer; (4) the Cr/Au film was etched, and the two Cr/Au electrodes were formed; (5) the electrode chip was immersed in the aqueous solution with an equal molar concentration (0.015M) of Zn(NO_3_)_2_ and HMTA at a temperature of 75 °C. At the same time, both the AC and CD electric fields were applied between the substrate and the Cr/Au electrodes using the method shown in Figure 1. During this process, the growing position and alignment of ZnO nanowires on the electrodes can be controlled by the applied electric fields; (6) after a period of time, the chip was taken out from the solution, rinsed with deionized water, and then dried in air. This fabrication process is also summarized in Figure 6.

### 2.3. Method of Growing ZnO Nanowires on PCBs

For the sake of reducing cost and facilitating fabrication, we attempted to use the electrodes deposited on printed circuit boards (PCBs) to replace the fabrication of microelectrodes, as shown in Figure 7. In this configuration, the dimensions of the two top electrodes were 2 mm × 2 mm, and their separation distance in the horizontal direction was 0.2 mm. The dimensions of the bottom electrode were 5 mm × 3 mm. The gap distance between the top and the bottom electrodes was 0.4 mm, which was guaranteed by the thickness of the PCB. The pads on the PCBs were used to provide connection points with an external power supply during the growth of the ZnO nanowires. After the fabrication of the PCBs through a normal industrial process, a thin layer of gold was deposited on the top electrodes. This gold layer provided the seeds for the growth of the ZnO nanowires. The same electric field application method given in Figure 1 was used here.

## 3. Results and Discussions

### 3.1. Requirement of Gas Discharge for ZnO Nanowires

In the discharger using ZnO nanowires shown in Figure 8, the electrode covered with ZnO nanowires served as the anode and was powered by a high voltage (HV). A grounded copper plate or an ITO glass was located dozens of micrometers apart from the anode. A Teflon spacer with a specific thickness was placed between the anode and the grounded cathode to form the discharge gap. When the electric field in the gap was sufficiently high, gas discharge took place, and large amounts of gas ions were produced.

In the needle-plate structures, high electrostatic fields were observed at the sharp points of the structures when the imposed voltage was sufficiently high. The purpose of using ZnO nanowires as anodes in gas discharge lies in the fact that they can provide very thin needles, which facilitate obtaining high electric field intensities with a relatively low voltage. In this scenario, only several hundred volts are needed to maintain the gas discharge. However, the electric field may be weakened next to nanowires if the distance between them is not sufficient. We numerically calculated the distribution of the electric field for discharge with one nanowire and two nanowires different distances apart, respectively. The results are shown in Figure 9. Typical nanowire and discharger dimensions were used. In this figure, the diameter and length of the nanowire were 240 nm and 1.5 μm, respectively. The distance and voltage difference between the anode and the cathode were 5 μm and 100 V, respectively. The maximum field intensity for each case is also listed in Table 2. When the nanowires were 0.26 μm apart, the field intensity at the tips of the nanowires decreased by 20% as compared to one nanowire. It is evident that the electric field at the tips of the nanowires for the dischargers with two nanowires weakened as compared with those with only one nanowire. This means that a 20% higher voltage is necessary to maintain continuous gas discharge due to the fact that the electric field intensity is proportional to the applied voltage.

The maintaining voltage for discharge is determined by the maximum electric field intensity in the discharger, which is usually located at the tips of the ZnO nanowire anodes. We simulated the field distribution for the cases with different distances between nanowires and extracted the maximum field intensity in each case for comparison. The results are given in Figure 10. It is evident that the maximum electric field intensity increased monotonously with the separation distance, but it did not attain the maximum with only one nanowire in the present calculated range. The progression change observed for the line representing the field intensity for only one nanowire indicates that a much larger distance between nanowires is needed to eliminate the impact of multiple nanowires. When the distance increased beyond approximately 4.5 μm, any further increase in the maximum field intensity almost vanished. As a result, we used a separation of approximately 4 μm to obtain the lowest maintaining voltage for gas discharge.

### 3.2. Growth of ZnO Nanowires on Microelectrodes

Experiments were conducted to grow ZnO nanowires on microelectrodes using the wet chemical method assisted by both AC and DC electric fields. For the sake of comparison, similar experiments were also carried out but were assisted with only AC or DC electric fields. Figure 11 shows the experimental results provided in the form of SEM images. The ZnO nanowires were grown on one pair of microelectrodes with an optimized shape, as shown in Figure 3.

For the cases in which only the AC field was applied (Figure 11(a1,a2)), the nanowire tips exhibited a dispersed distribution but grew only in the places between two electrodes. When this configuration was used in a discharger, some ion migration occurred in a small part of the discharge gap. This was not expected in a gas discharge application. For the cases in which only the DC field was applied (Figure 11(b1,b2)), the ZnO nanowires covered one whole electrode, but the distance between them was small, which severely weakened the electric field around the tips of the nanowires, based on the calculated results of the field distribution in Section 3.1. The site selectivity behavior may be attributed to the dielectrophoresis (DEP) force [26], which results in the nanowires gathering in the position with a larger field gradient. The field potential decreases towards the electrode ends in the cases in which a DC field is applied, but the field gradient is larger, close to the other ends of the two electrodes when an AC power is applied.

As regards the experimental results shown in Figure 11(c1,c2), both the AC and DC electric fields were applied. The voltages of the first and the second DC field (DC1 and DC2 in Figure 1) were −1 V and 0 V, respectively. The frequency and peak-to-peak voltage of the AC field were 1 MHz and 2 V, respectively. These parameters depend on the shape and dimensions of the microelectrodes, the values of which were determined in experimental trials. It can be seen that the ZnO nanowires covered the whole electrode as in the cases in which only the DC field was applied, and the tips of the nanowires were scattered, similar to what was observed under the application of the AC field only. The average distance between the tips of the highest nanowires was approximately equal to 4 μm, which is suitable for gas discharge, as discussed in Section 3.1.

### 3.3. Growth of ZnO Nanowires on PCBs

Experiments to assess growing ZnO nanowires on PCBs were conducted. Our results show that for the PCB configuration given in Section 2.3, the power supply parameters required to produce electric fields under which ZnO nanowires can grow are as follows: an AC field frequency of 1 MHz and a voltage of 0.4 V for the two DC fields. Several experiments were conducted under different peak-to-peak AC voltages, and the results are shown in Figure 12. For comparison, a growth experiment with only the DC field was also conducted. The statistical analysis shows that the average diameter of the ZnO nanowires increased from 140 nm to 284 nm when the AC voltage increased from 0.5 V to 2 V. The density of the distribution of ZnO nanowires in these experiments approached that observed using the DC electric-field-assisted method (shown in Figure 12d). The average distance between the nanowire tips in these experiments was equal to one order of magnitude of their diameter. Therefore, at present, the ZnO nanowires grown on PCB electrodes are not suitable for gas discharge anodes.

## 4. Conclusions

Using ZnO nanowires as needle anodes in gas discharges facilitates high electric field intensity with a relatively low voltage. As a result, a sufficient distance between ZnO nanowires is required to guarantee no electric field weakening. Another requirement is that the ZnO nanowires are easy to transfer to the discharging devices. An optimum shape of the microelectrodes was established by numerically calculating the electric field distribution when a voltage was applied. This shows that electrodes with less than six fingers obtain a relatively larger field intensity, and thus, this parameter was used in the experiments. As regards the typical dimensions of ZnO nanowires synthesized using wet chemical methods, the distance between the tips of ZnO nanowires should be greater than 4 μm approximately. An AC/DC electric-field-assisted wet chemical method was proposed herein. Furthermore, it was used to grow ZnO nanowires on microelectrodes and PCBs, respectively. For the experiments using microelectrodes, optimal dimensional parameters were determined by numerically calculating the electric field distribution in the dischargers. The experimental results show that the ZnO nanowires assisted by both AC and DC electric fields during growth covered the entire electrode, as was the case when only the DC field was applied. Moreover, the tips of the nanowires were scattered, similar to what was observed under the application of only the AC field. From the point of view of the average distance between the tips of the highest nanowires, the ZnO nanowires grown on microelectrodes almost meet the requirements of gas discharge, but those grown on PCBs did not. Further research is needed to establish the appropriate parameters in order to control the growing processes in experiments using PCBs.

## Figures and Tables

**Figure 1 materials-16-00108-f001:**
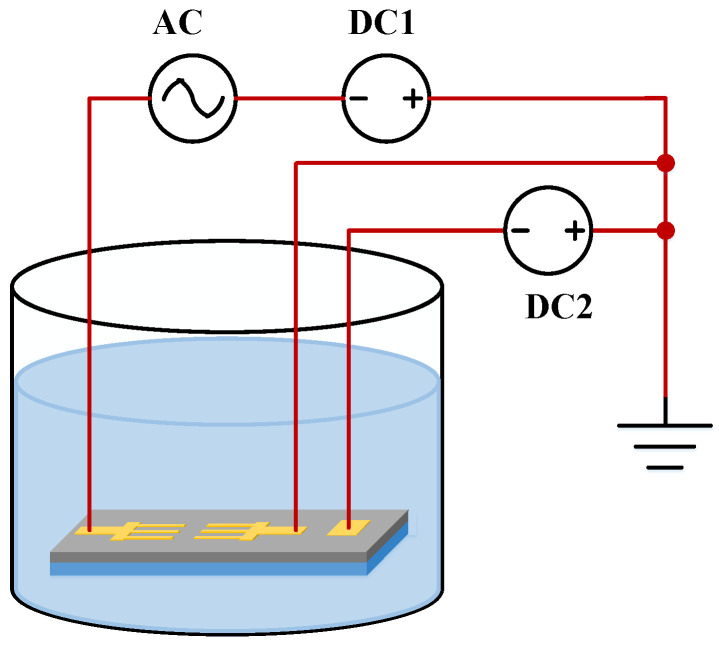
The electric field application method during the growth of ZnO nanowires.

**Figure 2 materials-16-00108-f002:**
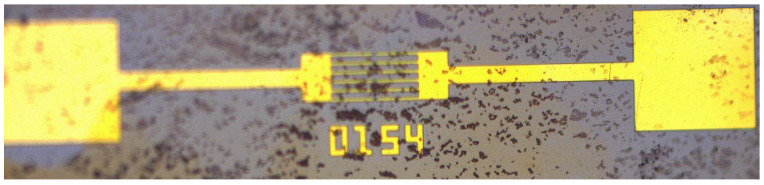
A picture of microelectrodes used for growing ZnO nanowires.

**Figure 3 materials-16-00108-f003:**
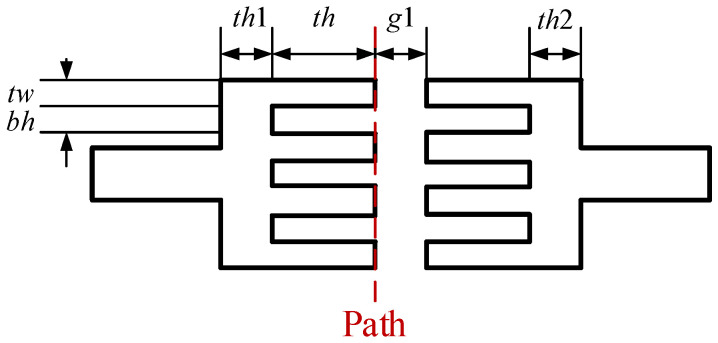
Dimensional parameters for controlling the shape of the microelectrodes.

**Figure 4 materials-16-00108-f004:**
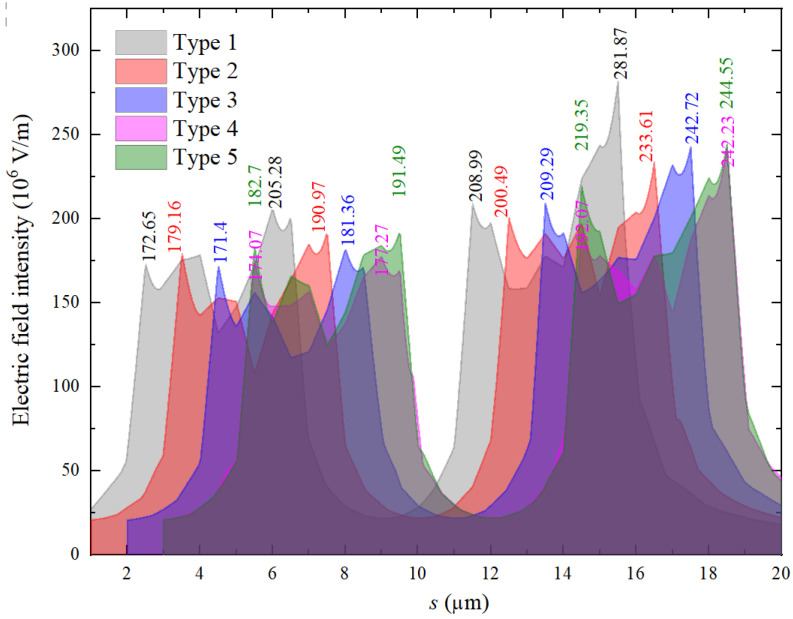
Distribution of the electric field around the fingers on the electrodes.

**Figure 5 materials-16-00108-f005:**
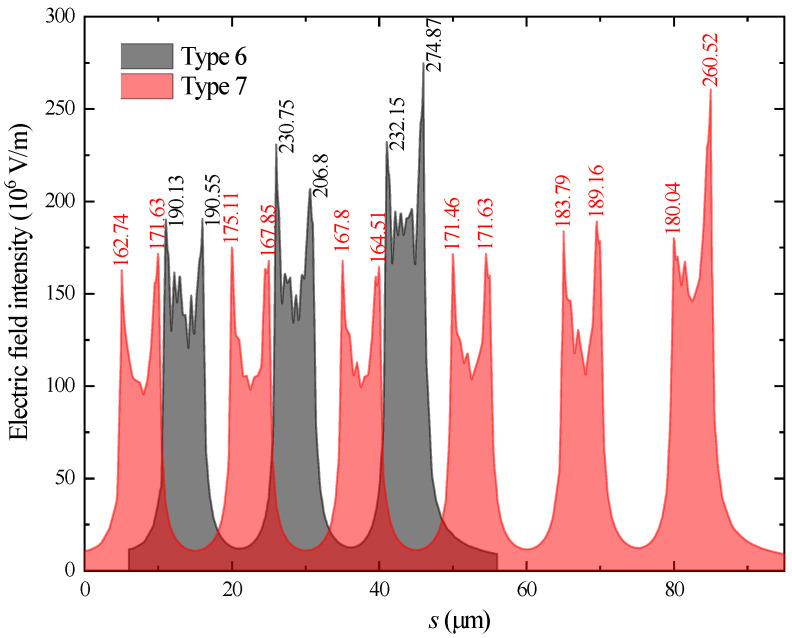
Distribution of the electric field around the fingers on the electrodes with 6 and 12 tips.

**Figure 6 materials-16-00108-f006:**
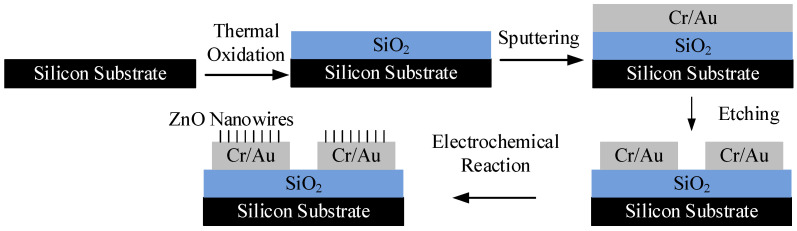
The fabrication process of ZnO nanowires on microelectrodes.

**Figure 7 materials-16-00108-f007:**
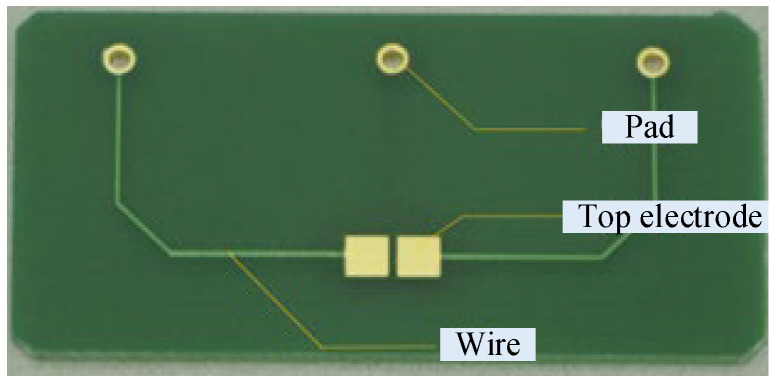
Electrodes on a PCB.

**Figure 8 materials-16-00108-f008:**
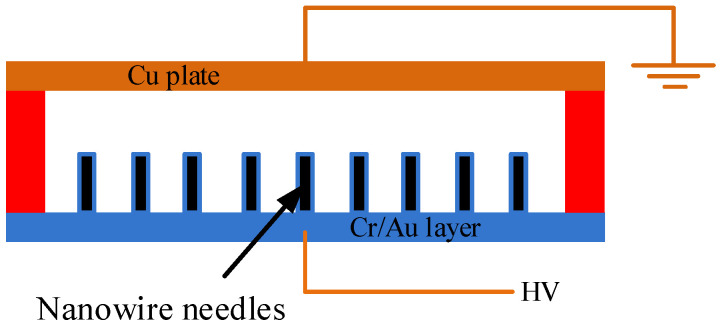
Schematic diagram of a discharger using nanowires as anodes.

**Figure 9 materials-16-00108-f009:**
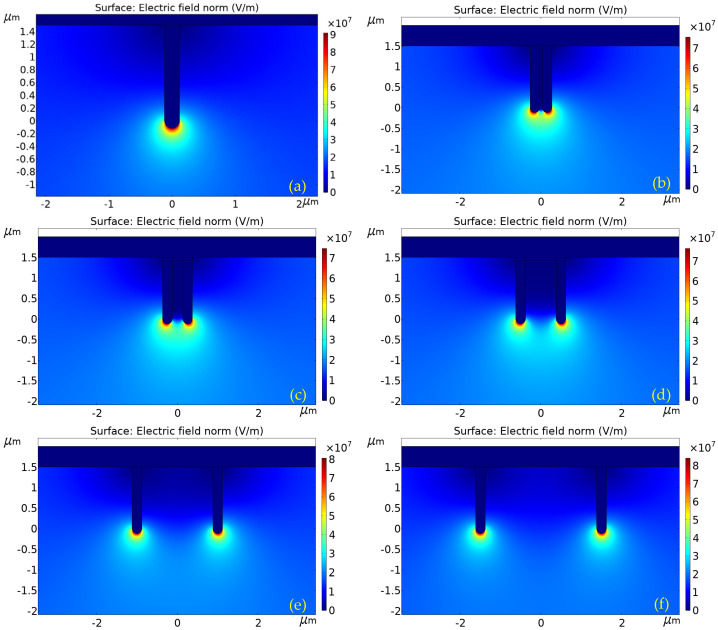
Distribution of electric field around the anode nanowires in the discharges. (**a**) One nanowire; (**b**) two nanowires, 0.06 μm spaced; (**c**) two nanowires, 0.26 μm spaced; (**d**) two nanowires, 0.76 μm spaced; (**e**) two nanowires, 1.76 μm spaced; and (**f**) two nanowires, 2.76 μm spaced.

**Figure 10 materials-16-00108-f010:**
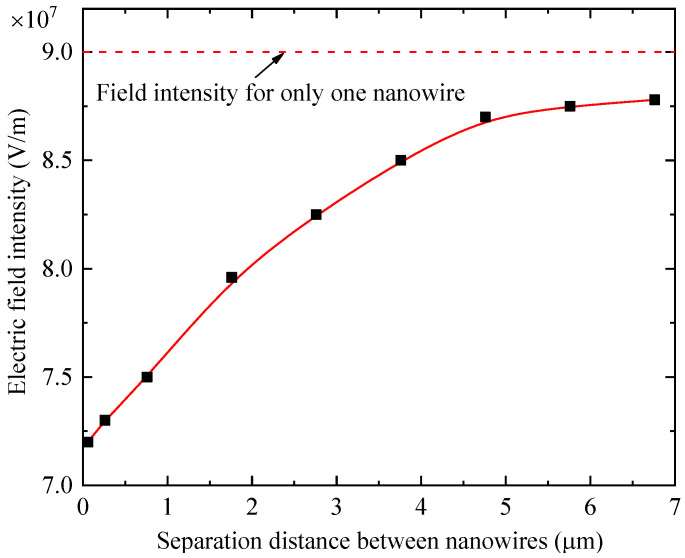
The maximum electric field intensity at the tips of nanowires in dischargers as a function of the separation distance between nanowires.

**Figure 11 materials-16-00108-f011:**
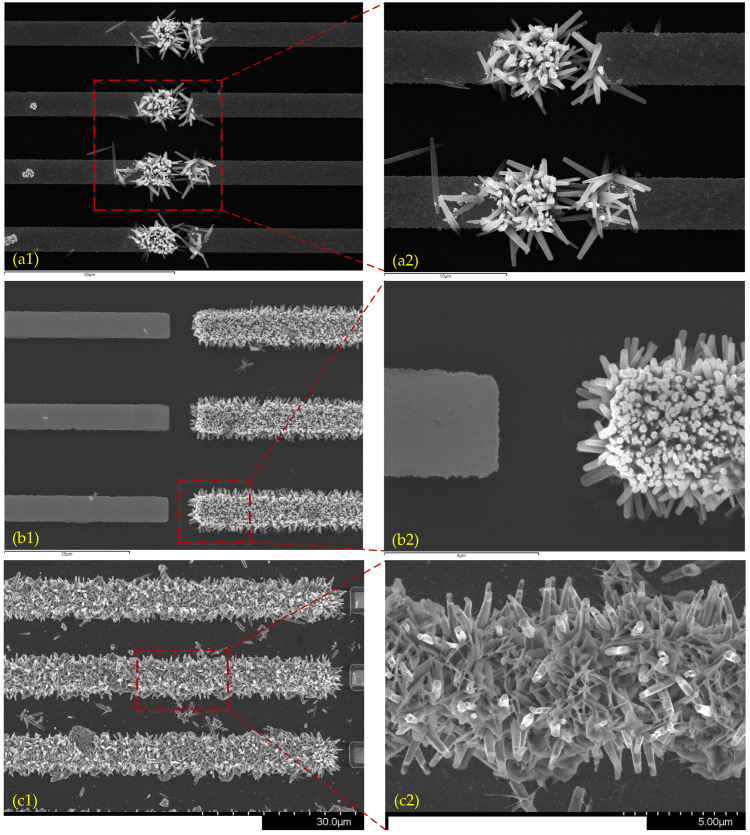
SEM images of ZnO nanowires grown on microelectrodes. (**a1**,**a2**) Only AC electric field is applied. (**b1**,**b2**) Only DC electric field is applied. (**c1**,**c2**) Both AC and DC electric fields are applied.

**Figure 12 materials-16-00108-f012:**
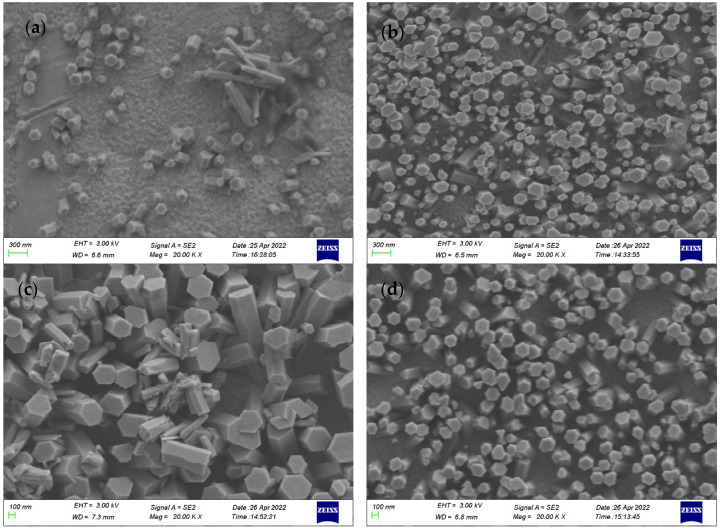
SEM images of ZnO nanowires grown on PCBs. (**a**) AC: 0.5 V, 1 Mz, DC1: 0.4 V, DC2: 0.4 V; (**b**) AC: 1 V, 1 Mz, DC1: 0.4 V, DC2: 0.4 V; (**c**) AC: 2 V, 1 Mz, DC1: 0.4 V, DC2: 0.4 V; (**d**) AC: 0 V, DC1: 0.4 V, DC2: 0.4 V.

**Table 1 materials-16-00108-t001:** Dimensional parameters of the microelectrodes.

Number	g1/μm	tw/μm	bh/μm	th/μm	Number of Tips
Type 1	1	4	5	20.5	4
Type 2	2	4	5	20	4
Type 3	3	4	5	19.5	4
Type 4	4	4	5	19	4
Type 5	5	4	5	18.5	4
Type 6	4	5	10	70	6
Type 7	4	5	10	70	12

**Table 2 materials-16-00108-t002:** The maximum electric field intensity $E_{\max}$ at the tips of the nanowires.

Number of Nanowires	Distance between Nanowires (μm)	$E_{max}$ (×10^7^ V/m)
1	-	9.08
2	0.06	7.21
2	0.26	7.26
2	0.76	7.51
2	1.76	7.96
2	2.76	8.25

## Data Availability

Not applicable.

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
