# Peer review of "AC/DC Electric-Field-Assisted Growth of ZnO Nanowires for Gas Discharge"

_materials, 2022, doi:10.3390/ma16010108_

Round 1

Reviewer 1 Report

Please refer the comments

Reviewer 2 Report

Comments to authors:

Language should be improved throughout the text – in some cases the language deficiencies makes it difficult to understand the meaning in the text.

Overall, the figure captions should be more descriptive

Quality/resolution of fig. 4 should be improved.

For figures 4 and 5 axes say “electric field intensity”, and captions say “magnetic field intensity”. Doesn’t seem right.

Line 166: Is a 250 nm  wide structure a nanowire? It is more like a mezzo/microwire.

Lines 186-189: Can authors explain, why further increase of the distance between doesn’t increase the resulting Electric field intensity?

The ordering of Figure a,b,c should be more like b,c,a, as b and c are mentioned before a in the text.

Lines 207-214: How were the AC and DC field values (voltage, frequency) selected?

Authors should provide an actual experiment not just a simulation of the acquired system (achievement of high electric field with low voltage).

Round 2

Reviewer 2 Report

0. Language still needs a lot of improvement.

1. Numbers in figure 4 are unreadable.

2. Lines 827-838. What should be the distance between the nanowires to achieve the field intensity of one nanowire? Is the model tested for such a case to validate it? I ask it, because from figure 10 it is not obvious, that the field intensity with 2 nanowires will ever reach the E of one nanowire.

3. Regarding lack of experimental data in the electrode's application in gas discharge anodes - this paper would look way better if there was even one experiment, showing the electric field intensity improvement using such an electrode geometry.
